# Characterization and Pathogenicity of Two Novel PRRSVs Recombined by NADC30-like and NADC34-like Strains in China

**DOI:** 10.3390/v14102174

**Published:** 2022-09-30

**Authors:** Yu Wu, Ouyang Peng, Qiuping Xu, Qunhui Li, Wei Li, Limiao Lin, Qingfeng Zhou, Xinbin Cai, Guangli Hu, Zuyong He, Yaosheng Chen, Hao Zhang

**Affiliations:** 1State Key Laboratory of Biocontrol, School of Life Sciences, Sun Yat-sen University, Guangzhou 510006, China; 2Guangdong Provincial Key Laboratory of Malignant Tumor Epigenetics and Gene Regulation, Sun Yat-sen Memorial Hospital, Sun Yat-sen University, Guangzhou 510006, China; 3Medical Research Center, Sun Yat-sen Memorial Hospital, Sun Yat-sen University, Guangzhou 510006, China; 4Wen’s Group Academy, Wen’s Foodstuffs Group Co., Ltd., Yunfu 527400, China

**Keywords:** PRRSV, phylogenetic analyses, recombination analyses, pathogenicity

## Abstract

Porcine reproductive and respiratory syndrome viruses (PRRSVs) pose a serious threat to the swine industry in China, which has caused great difficulties for porcine reproductive and respiratory syndrome (PRRS) immune prevention and control, due to its easily mutable and recombinant nature. In this study, two novel PRRSV strains, which were named GD-H1 and GD-F1, were isolated and fully sequenced from pig farms in Guangdong province, China. The phylogenetic analysis and recombination analysis revealed that the GD-H1 and GD-F1 were generated by the recombination of NADC30-like and NADC34-like strains which were different from the previously prevalent strain. Further pathogenic studies on piglets and sows found that the recombinant strains could cause piglets high fever, loss of appetite and lung lesions, but no piglets died. However, the recombinant strains could cause acute death and abortion in pregnant sow infection models together with average survival rates of 62.5% and 37.5% abortion rates, respectively. These findings indicated that the recombinant strains were extremely pathogenic to sows. Therefore, we report two clinical novel recombinant strains of PRRSV that are different from the traditional epidemic strains in China, which may provide early warning and support for PRRS immune prevention and control.

## 1. Introduction

PRRSV belongs to the family Arteryviridae, the genus Porartevirus, and it is an enveloped, positive-sense, single-stranded RNA virus [1]. The genome of PRRSV is approximately 15 kb and contains 10 open reading frames (ORFs), namely, ORF1ab, ORF2a, ORF2-7 and ORF5a [2,3]. The non-structural protein NSP2 encoded by ORF1a may affect viral addiction to cells or tissues [4]. ORF5 is a highly variable region, the structural protein GP5 that it encodes is one of the major antigenic proteins [5]. In general, based on the differences in geographic origin and genetic variation, PRRSV can be classified into type 1 (European genotype) and type 2 (North American genotype), represented by the Lelystad virus (LV) strain and the ATCC VR-2332 strain [2], respectively. In addition, PRRSV genotype 2 can be divided into nine lineages based on the ORF5 gene. Five lineal strains in China are CH-1a-like (8.7), VR2332-like (5.1), QYYZ-like (3.5), NADC30-like (1.8), and Western European type I isolates [6]. Recent studies demonstrate that new isolated NADC34-like PRRSV belongs to sublineage 1.5 and gradually becomes a dominant strain [7]. The coexistence of different lineages of PRRSV isolates in the field has attracted great attention to PRRSV recombination [8,9]. In particular, the NADC30-like strains characterized by the pattern of discontinuous 131 amino acid (aa) deletion in NSP2 are particularly prone to genetic recombination [10].

Porcine reproductive and respiratory syndrome (PRRS) is a highly contagious and mortality infectious disease characterized by respiratory disease in newborn piglets and swine reproduction disorders [11,12,13]. PRRSV was first reported in the United States in 1987 and then became an epidemic in Europe [11,14]. Since 1996, the strain CH-1a was first isolated in China [15], Especially since 2006 [16], the highly pathogenic PRRSV JXA1 strain caused massive economic losses to the pig industry and rapidly spread across the country [17]. In 2008, the NADC30 strain was first isolated in the United States [18,19]. Around 2013, new prevalent strains emerged in China, which were called NADC30-like PRRSVs due to their high homology with the US strain NADC30, such as JL580 [6,20]. In September 2018, another 1-7-4 PRRSV strain (FJ0908) [12] was isolated in Fujian province [4]. Then, the HLJDZD30-1902 strain belonging to NADC34-like isolates was found in Heilongjiang province [21,22]. Subsequently, a new type of PRRSV (NADC34-like PRRSV) appeared in China along with serious sow abortion [7,22].

In this paper, we isolated two novel PRRSV strains from piglets in Guandong province, China. Through whole genome sequencing analysis and recombination analysis, we indicated that those two strains were natural recombinant viruses by NADC34-like (sublineage 1.5) and NADC30-like (sublineage 1.8) isolates. Furthermore, the pathogenicity study in piglets and pregnant sows revealed that these two strains could cause the death of sows and the abortion of piglets. These results may provide important clues for studying the epidemic evolution of PRRSV and provide guidance for the prevention and control of PRRS.

## 2. Materials and Methods

### 2.1. Sampling and Virus Isolation

Since 2018, we have continuously collected a large number of clinical samples to detect the prevalence of PRRSV, such as tissue sample disposal. Dulbecco’s modified eagle medium (DMEM) supplemented with antibiotics was added to the clinical tissue samples for grinding, then stored at −80 °C, and subjected to three freeze–thaw cycles. The homogenate was centrifuged, and the supernatant was removed. Total RNA was extracted from the supernatant using an RNeasy kit (Magen, Shanghai, China). The PRRSV GP5 gene was detected by RT-PCR with the following primers: sense, 5′-ACCTGAGACCATGAGGTGGGC-3′; antisense, 5′-GCCAGAATGTACTTGCGGCCTA-3′. PRRSV positive samples were diluted with culture medium and then passed through 0.22 μm filters and inoculated onto primary alveolar macrophages (PAMs) for virus isolation. Nucleic acid testing and IFA testing were conducted for more than three consecutive generations to observe whether the separation was successful. Finally, two PRRSV strains were isolated and then identified by IFA testing and whole-genome sequencing, named GD-H1 and GD-F1, respectively.

### 2.2. RT-PCR Amplification and PRRSV Genome Sequencing

Total RNA was extracted from the cell culture samples using a RNeasy kit (Magen, Shanghai, China). All primers used to amplify PRRSV genomic fragments were designed and preserved in our laboratory. Reverse transcription and polymerase chain reaction (PCR) were performed using the PrimeScript RT Master Mix (TakaRa, Tokyo, Japan) and PrimeSTAR^®^GXL DNA polymerase (TakaRa, Tokyo, Japan), respectively. The PCR products were cloned into a pMD19-T vector using the TOPO^®^ TA Cloning^®^ Kit (Invitrogen, Carlsbad, CA, USA) and sent to Sangon Biotech for sequencing. For each amplicon, more than three independent clones were sequenced to determine the accurate sequence of a specific genomic region. Then, the whole viral genome was assembled.

### 2.3. Immunofluorescence Assay

PRRSV-infected or mock-infected cells were washed thrice with PBS, fixed at room temperature (RT) with 4% paraformaldehyde for 15 min, and then permeabilized with 0.2% Triton X-100 (Solarbio, Beijing, China) for 15 min. After washing as described previously, the cells were blocked with 3% BSA (Solarbio, Beijing, China) for 1 h at RT. Then, the cells were incubated with an anti-PRRSV N antibody (JN0401, Shanghai, China) overnight at 4 °C. Following three washes with PBS, the cells were incubated with anti-mouse IgG secondary antibody (Cell Signaling Technology, Boston, MA, USA) for 45 min at RT. All images were captured and processed utilizing an inverted fluorescence microscope.

### 2.4. Viremia and Serological Test

Viral RNA was extracted from the blood samples as described previously. The amount of viral RNA was determined to use the TaqMan real-time RT-qPCR with the following primers: sense, 5′-CTAGGCCGCAAGTACATYCTG-3′; antisense, 5′-TTCTGCCACCCAACACGA-3′. A probe targeting the PRRSV N gene (5′-FAM-TGATAACCACGCATTTGTCGTCCG-BHQ-3′) was also employed. The thermal cycling parameters were as follows: 95 °C for 20 s followed by 40 cycles at 95 °C for 3 s and 60 °C for 30 s.

RRSV-specific ELISA antibody titers were measured using Herdcheck PRRSV X3 antibody test (IDEXX Laboratories, Westbrook, ME, USA) as described by the manufacturer. PRRSV-specific antibody titer was reported as sample-to-positive (S/P) ratios. The serum samples with an S/P ratio of 0.4 or higher were considered positive.

### 2.5. Sequencing Results Analysis

All sequencing results were then spliced and aligned using Lasergene SeqMan and Lasergene MegAlign, respectively. All reference sequences were obtained from GenBank and used for sequence alignment and phylogenetic analysis. All phylogenetic trees were constructed using MEGA (v7.0) software.

### 2.6. Recombination Analysis

To test the recombination of GD-H1 and GD-F1, the multiple genome alignment was submitted to screen potential recombination events by recombination detection program 4 (RDP4). Seven methods embedded in the RDP4 software package, including RDP, GENECONV, BootScan, Maxchi, Chimaera, SiScan, and 3Seq, were utilized to detect recombination events and breakpoints. The detected recombination events were further confirmed by SimPlot 3.5.1, which was performed within a 200 bp window, sliding along the genome alignments with a 20 bp step size.

### 2.7. Animal Experiment Design

To assess the pathogenicity of the isolates GD-H1 and GD-F1, we performed challenge studies in piglets and sows. Fifteen healthy piglets aged 35 days and twelve pregnant sows were equally divided into six groups, respectively (five piglets per group and four pregnant sows per group). The challenges were performed by intramuscular injection of 2 mL with 10^5^ × TCID_50_ of GD-H1 or GD-F1 culture medium and 10 mL for sows, while pigs in the negative control were inoculated with an equal volume of RPMI-1640. Daily rectal temperature and clinical signs were recorded for each group. The clinical signs were scored from food intake, mental, skin, respiratory status, cough condition, runny nose, neurological symptoms, and stool condition, with each index scored from 0 to 5 points; the more obvious the disease, the higher the score [23]. Serum samples were collected, and the blood viral load and antibody levels were analyzed. For the piglets, the survival status was recorded during the experiment. Then, all piglets were euthanized 14 days post-infection (dpi), and the tissue samples including lungs, submandibular lymph nodes, and inguinal lymph nodes were collected for histopathology. For the pregnant sows, sow death, miscarriage, as well as delivery data were observed and recorded. For the daily feed intake measure, we gave the sows a certain amount of feed daily and then weighed the residual feed in the trough at a fixed time every day to calculate the daily feed intake of the sows.

### 2.8. Statistical Analysis and Data Visualization

In each experimental group, statistical significance was measured using one-way analysis of variance. Two-sided probability values < 0.05 (*p* < 0.05) were considered to indicate statistical significance.

All point plots were created using GraphPad Prism (v8.0.0) for Windows, GraphPad Software, San Diego, CA, USA, www.graphpad.com, accessed on 23 July 2022. Heatmaps were generated with TBTools (v1.092) [24]. Phylogenetic trees were visualized with R package ggtree (v3.0.4) [25] in R environment (v.3.5.1).

## 3. Results

### 3.1. Virus Isolation

In the past few years, we have continuously collected a large number of clinical samples, including fecal swabs, oral swabs, serum, tissue samples, piglet embryos, as well as environmental samples, where the maximum numbers of serum and tissue samples are 5789 and 3241, respectively. Meanwhile, the positive detection rate of PRRSV was the highest in tissue samples and piglet embryos, at 95.2% and 97.9%, respectively (Appendix A). Recently, we isolated and identified two similar strains from two different pig farms with severe PRRS outbreaks in Guangdong province, China (Figure 1A). Firstly, the PRRSV GP5 gene (603 bp) fragment was amplified to screen for the positive samples (Figure 1B), then the PRRSV positive samples were utilized for further virus isolation using PAMs. Finally, we obtained two PRRSV strains on PAMs cells through the process of successive generations of culture, and then their biological characteristics were evaluated. With the time of infection increasing, cellular lesions became more obvious, such as cell death and shedding (Figure 1C). Meanwhile, the IFA results showed that the fluorescence positive signal increased with the culture time increasing (Figure 1D). In addition, the detection of the virus titer and nucleic acid proliferation revealed that both strains’ growth tendency was of high similarity (Figure 1E,F). In conclusion, through isolation and identification, we obtained two PRRSV strains named: GD-H1 and GD-F1, respectively.

### 3.2. Genomic Characterization

The genome lengths of GD-H1 and GD-F1 were both 15,019 nt, excluding the 3′-end poly (A) tail. The complete genome sequences of the two strains described here have been deposited in GenBank under accession no. ON691479 and no. ON691480. Seven representative reference strains of PRRSV in different lineages were downloaded from NCBI to analyze the nucleotide sequence similarity of GD-H1 and GD-F1. Full-length genome sequenc alignment revealed that GD-H1 and GD-F1 shared 86.3% sequence identity with NADC34 (sublineage 1.8), 86.1% with IA/2015/ISU-10 (sublineage 1.8), 90.0% with NADC30 (sublineage 1.5), 84.6% with VR2332 (sublineage 5.1), 82.2% with QYYZ (lineage 3), 84.7% with CH-1a (sublineage 8.1), and 84.5% with JXA1 (sublineage 8.3) (Appendix A) (Figure 2A). These data suggested that PRRSV GD-H1 and GD-F1 shared the highest similarity with NADC30-like strains among all PRRSV sublineages in full-length genome. Furthermore, we compared the nucleotide homologies of the ORF1a, ORF1b, and ORF2-7 genes of the GD-H1 and GD-F1 strains with the other reference strains. The results showed that the ORF1a and ORF1b gene segments of the GD-H1 and GD-F1 strains showed higher nucleotide identity to NADC30-like PRRSV than the other PRRSV strains, from 88.2–93.2% and 88.1–93.7%, respectively. All other gene segments have higher homology with NADC34-like PRRSV; especially in the ORF5 gene segment, the similarity is 96%, while the nucleotide identity to NADC30-like PRRSV is only 87.6–86.9% (Appendix A).

In addition, using SimPlot 3.5.1, we analyzed the similarity between two isolated strains with the classical strains VR-2332, NADC30, and NADC34. The results showed that the two strains had a low similarity with VR-2332, especially at positions 0 to 5000 bp, which was only about 50% (Figure 2B). For NADC30 strains, both strains showed higher similarity, but at nucleotide position 12,500, there occurred a low similarity (Figure 2C). Comparing to the NADC34 strain showed low similarity around nucleotide 2500, but high similarity at nucleotides 12,500 to 15,000 (Figure 2D). In brief, the software similarity analysis showed consistency with the homology results.

### 3.3. NSP2 Amino Acid Analysis

NSP2 is the most variable protein in all PRRSV NSPs, responsible for the genetic evolution and pathogenicity of the virus. A comparison of nucleotide homology showed that GD-H1 and GD-F1 have a higher nucleotide identity with 89.2–89.2% NADC30-like PRRSV (Appendix A). To investigate the mutation profile, we analyzed and compared the NSP2 amino acid sequences of GD-H1 and GD-F1 with other PRRSV reference strains. The results showed that the “111 + 1 + 19” amino acid deletion of the NSP2 gene occurred in the same way as the molecular mutation of NADC30-like strains, which further indicated that GD-H1 and GD-F1 strains are recombinant strains (Figure 3).

### 3.4. Phylogenetic Analysis

For further analysis of GD-H1 and GD-F1 genetic evolution, we constructed phylogenetic trees based on the ORF5 gene and full-length genome sequences with the certain 61 strains (Appendix A). Phylogenetic analysis showed that based on the full-length genome, the GD-H1 and GD-F1 strains were clustered with NADC30-like (sublineage 1.8) (Figure 4A). However, phylogenetic analysis of ORF5 showed that the two strains were clustered with NADC34-like (sublineage 1.5) (Figure 4B). Genetic evolution analysis of ORF5 determines the classification of the viral subpopulation and is a variable gene widely used for phylogenetic analysis. At present, the popularity of genotype 2 PRRSV, found in China, is classified into lines 1, 3, 5, and 8 based on the genetic diversity of the ORF5 genes. In addition, we performed genetic evolution analysis based on other ORFs. The results showed that these two isolated strains were clustered with NADC34-like strains based on ORF2, 3, and 4, while for ORF1a, 1b, and 7, two strains were clustered with NADC30-like strains (Appendix A).

### 3.5. Recombination Analysis

Multiple alignment analysis based on the representative PRRSV genome was performed with RDP4 software. The results showed that the seven methods in RDP4 strongly supported that GD-H1 and GD-F1 were natural recombinant viruses of NADC30-like and NADC34-like isolates. For GD-H1, NADC30 is the major parental virus while HLJZD30-1902 is the minor parental virus. Two gene fragments occurred during the recombination at 11,722–12,778 and 13,471–14,033 bp, respectively (Appendix A). For GD-F1, NADC30 is the major parental virus while NADC34 is the minor parental virus (Appendix A). Three recombinant gene fragments happened at 6124–6535, 11,688–12,778 and 13,305–14,614 bp, respectively. Moreover, the multiple crossover events were further confirmed using SimPlot3.5.1 (Figure 4). Taken together, we isolated two novel recombinant PRRSV strains of GD-H1 and GD-F1 from the clinical samples, which not only had the characteristics of NADC30-like but also had that of NADC34-like, strongly supporting the RDP4 results (Figure 5) [26].

### 3.6. Pathogenicity Analysis

The pathogenicity results data showed a gradual increase in body temperature on the third and fourth days after the infection, reaching the highest temperature at 9 dpi, after which the body temperature gradually returned to normal (Figure 6A), while two strains had higher body temperatures than the control group throughout the experiment, causing fever in the pigs. In terms of clinical symptom scores, the onset was more pronounced in the GD-H1 and GD-F1 groups compared to the control group, with the highest score at 10 dpi (Figure 6B). In addition, the viral RNA load of each group was tested after challenge, and it was found that high levels of viral load were detected in the blood, oral swabs, and fecal swabs from 3 dpi, indicating that the two strains were not only able to cause high levels of viremia but also shedding through the oral and fecal routes (Figure 6C–E). We also tested the serum for PRRSV N antibodies and found that two strains caused positive PRRSV antibody transitions and produced high levels of antibodies by 14 dpi (Figure 6G). Finally, the weight of each group was compared before and after the infection and it was found that there was no difference in weight gain between the two strains, while the control group showed a significant increase in weight gain (Figure 6F).

No mortality occurred in any of the PRRSV challenge groups during the experiment. At 14 dpi, pigs in each group were euthanized. Lungs, submandibular lymph nodes, and inguinal lymph nodes were observed and collected. At necropsy, two strains could cause local hemorrhage in the lungs, as well as hemorrhage and atrophy in the submandibular and inguinal lymph nodes (Figure 7A). Further histopathology tests revealed damage to the alveoli after infection, resulting in inflammatory infiltration and interval thickening in all the infection groups and none in the control group (Figure 7B). Meanwhile, the bleeding site was also found in lymphoid histopathological sections shown by the white arrow (Figure 7A).

### 3.7. Pregnant Sows Pathogenicity Analysis

Furthermore, two recombinant strains’ pathogenicity was assessed by the challenge of pregnant sows. This experiment lasted for 42 days, during which the sows were challenged and then delivered. The body temperature of the pregnant sows in the challenge groups increased after the challenge, which was maintained for about a week, and then returned to normal (Figure 8B). At the same time, the clinical symptom score also showed a high value after attack (Figure 8D), and the challenged sows showed a significantly reduced feed intake (Figure 8C). In addition, the GD-H1 group had one sow miscarriage at 4 and 18 dpi respectively, and one sow died at 6 dpi, while one sow in the GD-F1 group died at 10 and 15 dpi, respectively (Appendix A). The final sow delivery data showed that the GD-H1 group sows had a 75% survival rate and a 50% miscarriage rate; only one sow farrowed, with 13 piglets delivered, but only two weak piglets survived, and the rest were mummified pigs (Appendix A) (Figure 8A). Sows in the GD-F1 group had a 50% survival rate and a 25% abortion rate (Appendix A). The remaining one sow delivered 14 piglets, but none of them survived (Appendix A) (Figure 8A). For the control group, there was no sow abortion or death during the whole experiment, and the farrowing was normal, with no obvious stillbirth or obvious weak piglets (Figure 8A). Finally, the antibody level of the sows after the challenge was tested. Starting from 5 dpi, the PRRSV-N antibody in the GD-H1 and GD-F1 groups gradually increased, while the control group was negative (Figure 8E).

## 4. Discussion

In China, PRRSV was first reported in 1996, with its pathogen-isolated strain named CH-1a [27]. Between 1995 and 2005, PRRSV was mainly locally endemic. In the summer of 2006, a novel PRRSV variant strain, also known as the highly pathogenic PRRSV (HP-PRRSV), caused atypical PRRS with devastating damage to the pig industry, producing a lethal rate of at least 20%. Subsequently, the HP-PRRSV-like strains became a major epidemic strain in pig farms [28]. Since 2013, the NADC30-like strains have undergone extensive recombinant variation and prevalence in China [8], resulting in the poor cross-protection of the strains with the current vaccines. The NADC34-like strain was first identified in Liaoning province in 2017 [22], causing the miscarriage and death of sows and posing a new threat to production. Thus, we have continuously collected a large number of clinical samples to detect the prevalence of PRRSV in the past few years [29]. Recently, we isolated two mutant strains from two different pig farms in Guangdong province, named GD-H1 and GD-F1, which were different from the existing strains.

GD-H1 and GD-F1 showed the highest nucleotide identity with NADC30 at 90.2% and 90.0%, respectively, in the perspective of whole genome analysis. However, based on the ORF5 gene, GD-H1 and GD-F1 showed the highest homology with NADC34, both at 96%. These data were consistent with the phylogenetic analysis results based on the full-length genome and ORF5. It was known that the main structural protein GP5 encoded by ORF5 is involved in the formation of the PRRSV envelope [26]. GP5 is also proposed to be the primary target for neutralization, of which importance is supported by the identification of a neutralizing epitope and the discovery of neutralization-resistant mutants in GP5 [12,26,30,31]. Therefore, the genetic analysis, including the nucleotide acid mutations and fragments recombination of ORF5, portends a potential evolutionary direction of PRRSV [6,32]. The two novel strains GD-H1 and GD-F1 in our study both have recombinant fragments derived from NADC30 and NADC34-like strains, suggesting that it may lead to the immune escape of the PRRSV and bring more difficulties to the prevention and control of PRRS.

The gene encoded NSP2 is one of the most variable regions in the whole PRRSV genome. The deletion, insertion or mutation of the NSP2 gene could be used as an ideal molecular marker [28]. As a molecular marker, HP-PRRSV in China showed discontinuous 30 aa deletion (1 aa + 29 aa) in NSP2 [33]. The amino acid arrangement of the NADC34-like strain NSP2 protein had 100 aa deletion at 328–427 [21]. Our data showed that the NSP2 genes of isolates GD-H1 and GD-F1 exhibit a “111 + 1 + 19” amino acid deletion pattern, which was in the same way as the molecular mutation of NADC30-like strain [34]. In addition, NSP2 INDELs characterizing discontinuous 75 amino acids (from position 476 to 552 aa) or continuous 120 amino acids (from position 628 to 747 aa) deletion of NADC30-like strains were constantly identified [35,36], indicating a high complexity of the ongoing PRRSV epidemic in the field. In consideration of the high variability of the PRRSV NSP2 gene, these INDELs certainly increase the genetic diversity of the PRRSV and continuously endanger the development of the pig industry.

It is very common that recombination events occur in different PRRSV isolates in pig farms, especially for the NADC30-like strains. Such recombination events between HP-PRRSV and NADC30-like PRRSV, NADC30-like strains recombined with VR-2332-like and CH-1a-like classical strains has also been identified in recent years [28]. In January 2022, Harbin Animal Research also reported the emergence of similar recombinant strains [29]. Moreover, several pathogenicity studies have shown that NADC30-like viruses isolated in recent years, such as CHsx1401, FJ1402 and HN201605, can cause fever and respiratory symptoms but no mortality [37,38,39]. Additionally it has also been shown that NADC30-like can cause interstitial pneumonia, hemorrhagic necrotizing nephritis, and varying degrees of hemorrhage and atrophy in systemic lymph nodes in piglets, such as FJZ03 [40]. In our study, we found that the recombinant strains could cause pig fever, loss of appetite, 100% morbidity, and lung hemorrhage with multiple lymph nodes in piglets, but no piglets died, which was similar to the pathogenicity of the NADC30-like strain.

NADC34-like PRRSV was first reported in Liaoning province, China, in 2017 [41,42]. Studies have shown that different recombinant NADC34 causes different loss rates, such as the sow abortion rate ranging from 0–25% and the piglet mortality rate ranging from 10–80%, with the strain named PRRSV-ZDXYL-China-2018-2 originating from Heilongjiang, which can cause 80% piglet mortality [43]. For our study, we devaluated the pathogenicity of two isolated strains in pregnant sows. The results showed that both strains could cause acute sow death and abortion, except for causing fever and a significant decline in feed intake. The final sow delivery data showed that the GD-H1 group sows had a 75% survival rate and a 50% miscarriage rate; only one sow farrowed, with 13 piglets delivered, but only two weak piglets survived, and the rest were mummified pigs. The sows in the GD-F1 group had a 50% survival rate and a 25% abortion rate. The remaining one sow delivered 14 piglets, but none of them survived. These were similar to the NADC34-like PRRSV pathogenicity [22,29,42].

Since the development of the PRRSV epidemic in China, the evolution, mutation, and recombination of PRRSV strains has severely threatened pig breeding. Compared to the current treatment and prevention measures of PRRSV, the traditional vaccine has a weak cross-protection against the strains, and early warning and prevention are effective control measures. Furthermore, there is a great possibility that a new recombinant strain will emerge between the two sublineages, NADC30-like and NADC34-like strains, as they have been very common in recent years [18,29]. At the beginning of 2022, the Harbin Veterinary Research Institute reported the occurrence of NADC30-like and NADC34-like recombinant strains in pig farms; however, their pathogenicity is yet to be explored [29]. Our study shows that NADC30-like strains are less virulent in piglets and sows, while NADC34-like strains are less pathogenic to piglets, and the severity of the pathogenicity to sows has been suggested to be due to the recombinant strain [44]. However, further clinical trials with animal models are required to test whether the recombination of these two strains results in enhanced pathogenicity. Accordingly, in our animal challenge experiments, the recombinant strains were highly pathogenic to sows and directly lead to the death and abortion of sows. Our preliminary indications suggest that the epidemic of such recombinant strains may present a serious threat to the pig industry in the future.

In conclusion, we isolated and reported two novel strains recombined by NADC34-like and NADC30-like isolates, named GD-H1 and GD-F1. Meanwhile, the pathogenic experimental study of the recombinant strains both in piglets and sows found that these two strains are slightly pathogenic to piglets, but highly pathogenic to pregnant sows, which can lead to the acute death and abortion of sows, indicating that the emergence and prevalence of such strains may bring new threats and challenges to pig breeding.

## Figures and Tables

**Figure 1 viruses-14-02174-f001:**
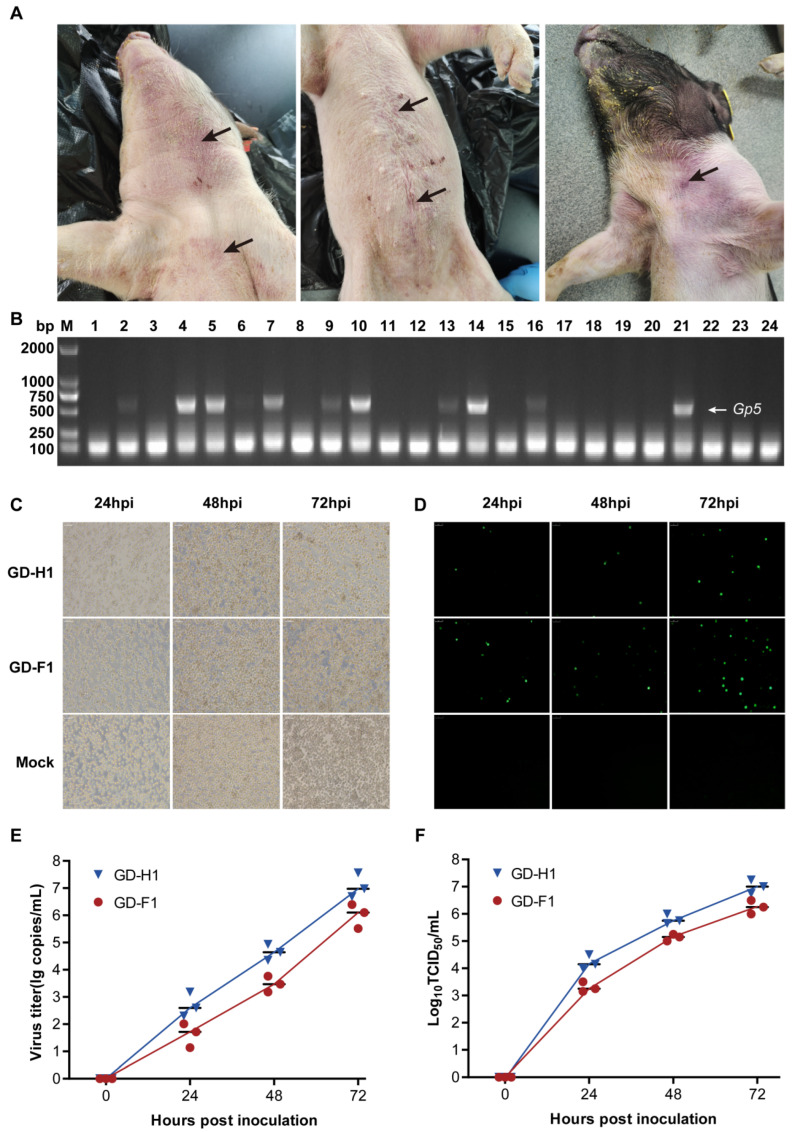
Strain isolation and identification. (**A**) Clinical symptoms of PRRSV-infected pigs; black arrows mark the lesion sections. (**B**) Clinical samples were examined by gel electrophoresis. (**C**) Cytopathic conditions at 24, 36, and 72 h post-infection (hpi). (**D**) IFAs showing the reactivity of a monoclonal antibody against PRRSV N protein to GD-H1 and GD-F1 strains infected at 24, 36, and 72 hpi. Magnification, 400×. (**E**) Viral titer detection of GD-H1 and GD-F1 strains at 24, 36, and 72 hpi. (**F**) Growth kinetic of GD-H1 and GD-F1 strains.

**Figure 2 viruses-14-02174-f002:**
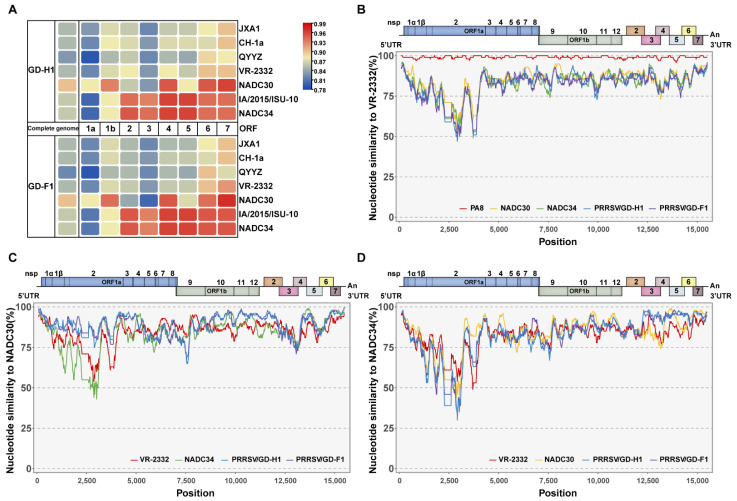
GD-H1 and GD-F1 strains’ amino acid similarity to classical strains. (**A**) heatmap compared the similarities of isolated strains and other reference strains. (**B**–**D**) Comparison of GD-H1 and GD-F1 strains’ similarity to VR-2332, NADC30, and NADC34 strains.

**Figure 3 viruses-14-02174-f003:**
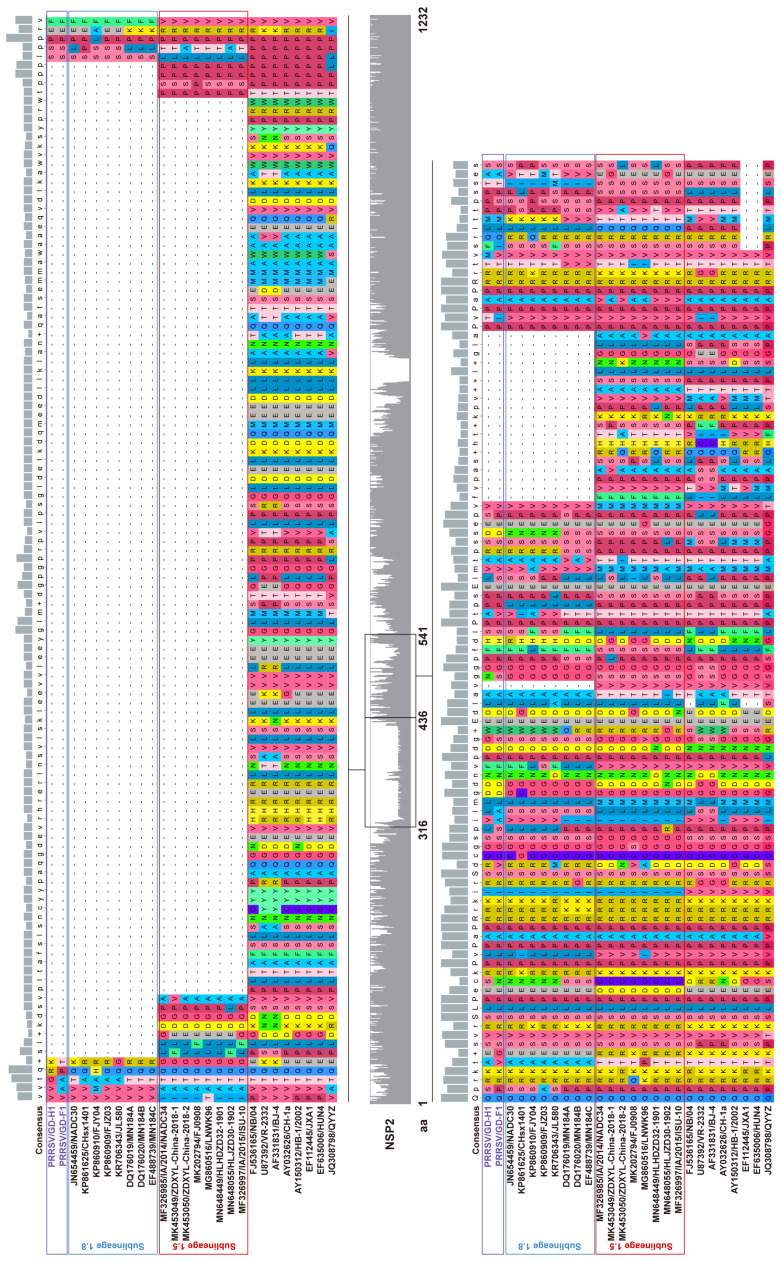
Alignment of the deduced amino acid sequence based on the NSP2 gene. The alignment of the GD-H1 and GD-F1 strains is marked with purple.

**Figure 4 viruses-14-02174-f004:**
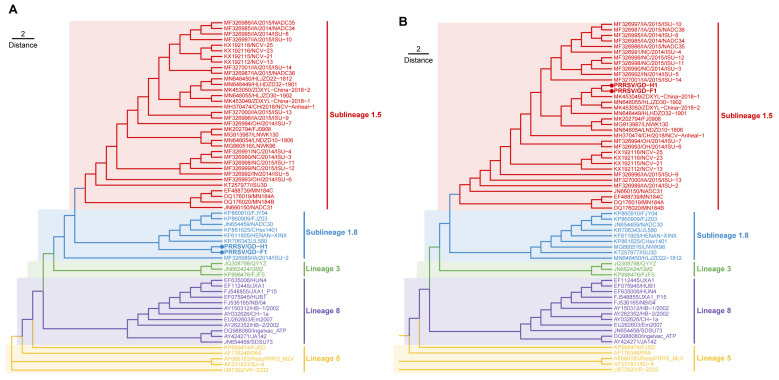
Phylogenetic and recombination analysis of GD-H1 and GD-F1 strains. (**A**) Phylogenetic trees constructed based on the ORF5 gene of GD-H1 and GD-F1 strains with 61 reference PRRSV strains. (**B**) Phylogenetic trees constructed based on the full-length genomes of the GD-H1 and GD-F1 strains with 61 reference PRRSV strains. The GD-H1 and GD-F1 strains are marked with a round dot. The NADC34-like and NADC30-like isolates are labeled in red and blue.

**Figure 5 viruses-14-02174-f005:**
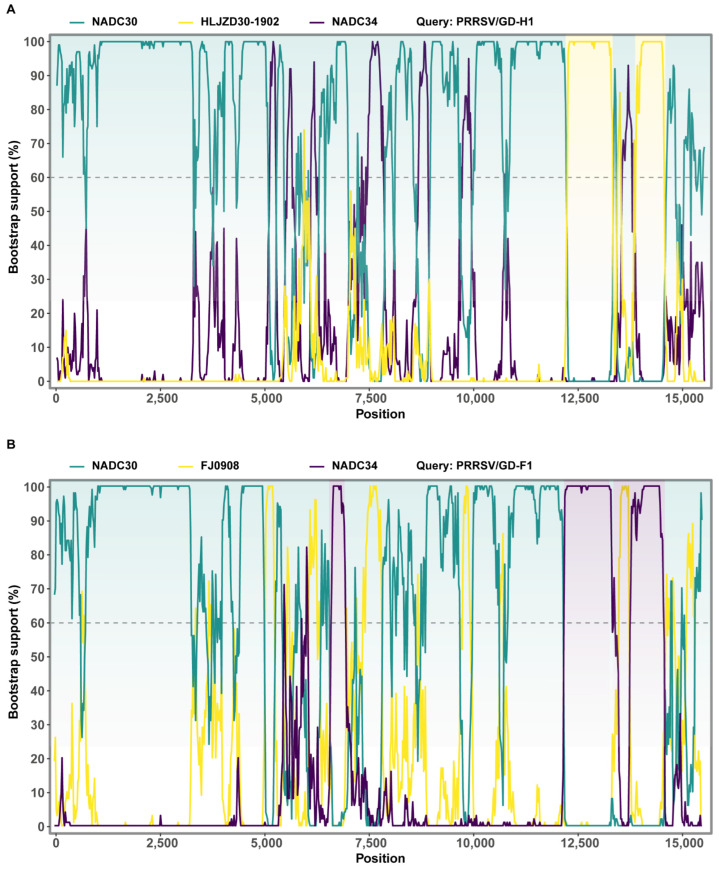
The crossover regions in the GD-H1 and GD-F1 genomes were further confirmed by SimPlot 3.5.1. (**A**) For GD-H1, NADC30 is the major parental virus while HLJZD30-1902 as the minor parental viruses. (**B**) For GD-F1, NADC30 is the major parental virus while NADC34 as the minor parental viruses. The crossover regions identified by SimPlot were consistent with the results from RDP4 analysis (Appendix A). The y-axis shows the percentage of permutated trees employing a sliding window of 200 nucleotides (nt) and a step size of 20 nt.

**Figure 6 viruses-14-02174-f006:**
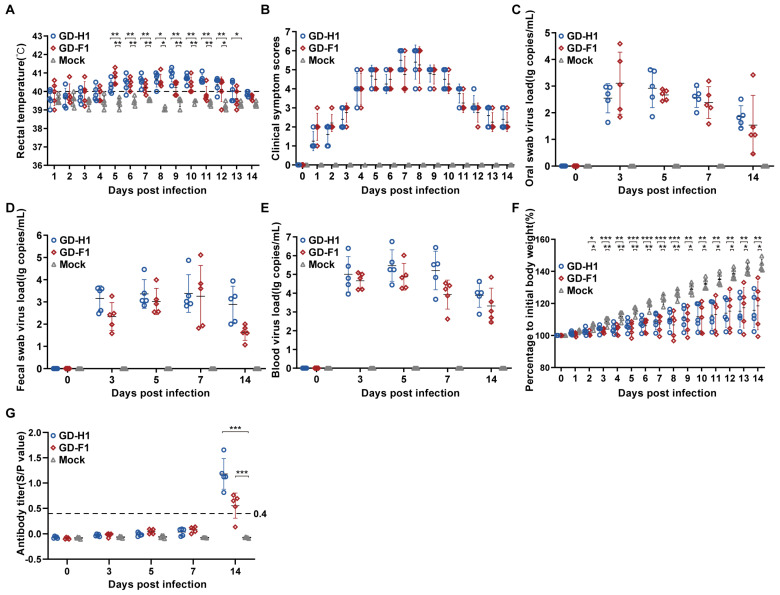
Pathogenicity results in piglets. (**A**) Body temperature change of pigs in each group after challenge. (**B**) Clinical scores of pigs after challenge during the entire experiment. (**C**–**E**) Viral load detection in blood, oral swabs and pharyngeal swabs. (**F**) Body weight gains of each group during the challenge study. (**G**) PRRSV-specific antibody level was detected in each group during the challenge study. Each bar represents the average for all pigs in each group ± standard deviation (SD). The significant difference is marked with the asterisk, *** *p* < 0.001, ** *p* < 0.01, and * *p* < 0.05.

**Figure 7 viruses-14-02174-f007:**
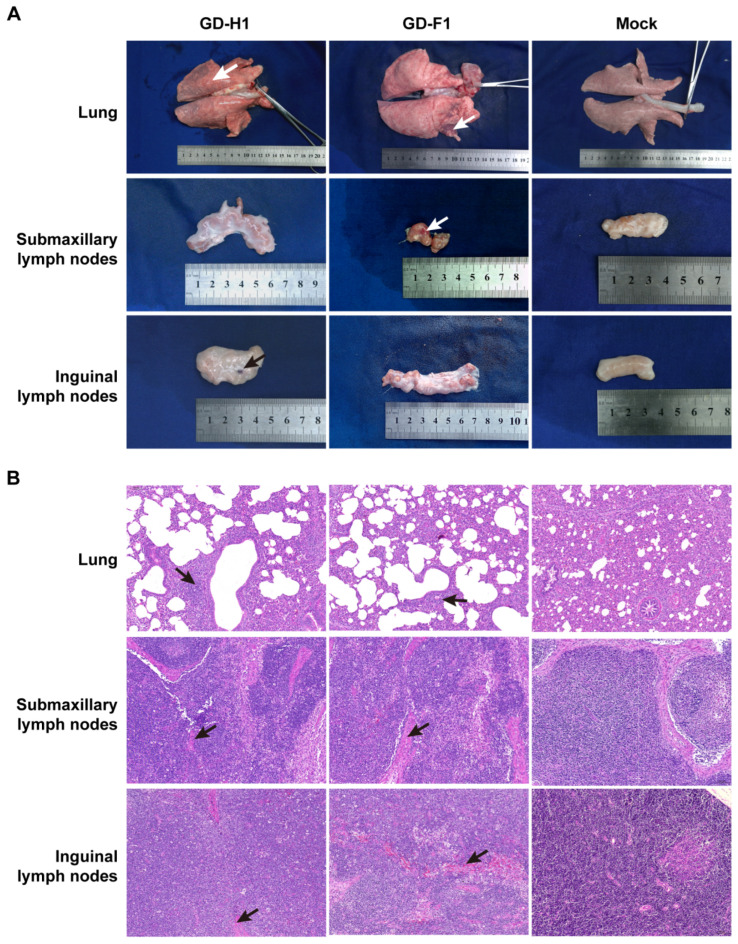
Observation and detection by pathological necropsy. (**A**) Necropsy observation of lung, submandibular lymph nodes and inguinal lymph nodes. White arrows mark the site of tissue bleeding. (**B**) Histopathology tests of the lung, submandibular lymph nodes and inguinal lymph nodes. Black arrows mark the lung lesions site in HE straining sections, as well as lymph node bleeding lesions.

**Figure 8 viruses-14-02174-f008:**
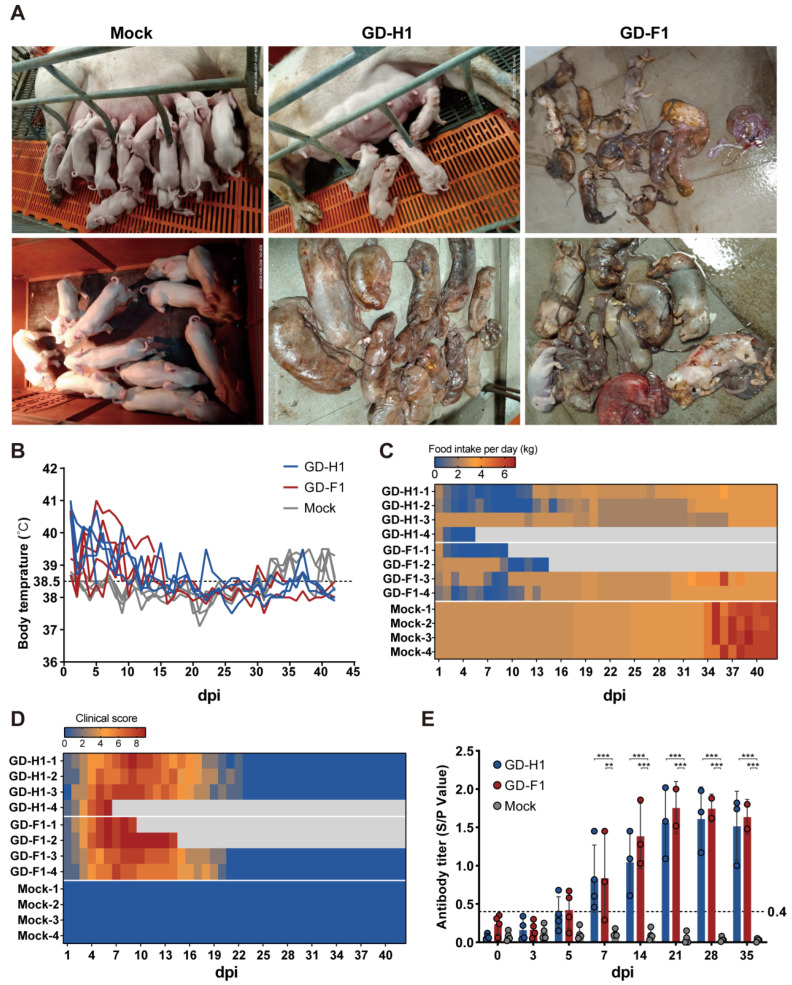
Pathogenicity results in pregnant sows. (**A**) Clinical photographs of the sow farrowing. (**B**) Body temperature change of pregnant sows in each group after challenge. (**C**) Food intake changes of pregnant sows in each group after challenge. (**D**) Clinical scores of pregnant sows after challenge during the entire experiment. (**E**) PRRSV-specific antibody level was detected in each group during the challenge study. Each bar represents the average for all pigs in each group ± standard deviation (SD). The significant difference is marked with the asterisk, *** *p* < 0.001 and ** *p* < 0.01.

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
