# Peer review of "Characterization and Pathogenicity of Two Novel PRRSVs Recombined by NADC30-like and NADC34-like Strains in China"

_viruses, 2022, doi:10.3390/v14102174_

Round 1
Reviewer 1 Report
The authors described the characterization and pathogenicity of two novel PRRSVs found in China.The topic was appreciated, the manuscript was well written, and the section well discussed. The work will benefit from minor revisions to correct some minor errors or aspects and provide some information. My comments are as follows:
Line 33: Authors wrote “contact”. Is it “Contagious”?
Line 36: I think you must provide a verb before “endemic” (become, be, etc.)
Line 46-54: I think that this part may be moved before the information of different strains. (line 36)
Line 82: Please, authors should briefly indicate how IFA was performed and which antibody was used. Being described in the following subparagraph, you could just write “described below”.
Line 86: Please, provide the primers used in the study and eventual appropriate reference.
Line 149: What the authors mean for “Cytoplesions”? Is it “Cytophatic effect”?
Fig.1: Graphs, please change “incoultion” in “inoculation”.
Discussion: Since GDH1 appeared to be more efficient in infecting the PAMs in vitro and in causing more relevant symptoms in infected animals, this result should be specifically discussed.
Reviewer 2 Report
This manuscript provides new information concerning the characterization and pathogenicity of 2 PRRSVs recombinant strains. The study seems well-designed and implemented. The authors made sequence analysis and viral challenge experiments to support their view. The study can value the current pig production by deciphering the PRRSV evolution and epidemiology in China.
My major concerns to this work are that 1) the manuscript is poorly prepared. The text lacks clarity in Material and Methods section, Results, Tables and Figure Legends. Moreover, a good English editing is suggested. My view is that the manuscript is not yet ready for publication in its present form. 2) The significance of PRRSV recombination between NADC30 and NADC34 should be analyzed and discussed. Does such recombination has been found elsewhere/ or it is the first report here? How this recombination made the changes to the infectivity or pathogenicity of current prevalent PRRSV strains? How this recombinant strain affects the current pig production and how we should act to the emergence of new recombinant strains?
Furthernore, below the Minor Points that need to be addressed by the Authors.
1) Line 88, PrimeScript™ RT-PCR kit does not belong to Promega products.
2) Lines 99-100, antibody information should be provided.
3) Line 149, what is “cytoplesions”?
4) Figure 1A, please describe the detailed macro- or microscopic manifestations of the shown lesion sections, as the specific arrows were used to point to some parts which actually cannot be visually distinguished the adjacent tissues in your images.
5) Figure 1B, what do you mean by using “gel electrophoresis” here? I guess it is a PCR result. I cannot find any information about what gene or sequence you used for amplification and how you determine which samples are positive from this result.
6) Figures 1C&D, high resolution images for cell morphology and staining should be provided. The cells and fluorescence shown in fig 1C&D was barely discernible in the reviewed version of manuscript.
7) Figures 1E&F, all data points are not placed at the indicated time points.
8) Line 140, please describe what is “diseased material”.
9) Table 1 should be placed prior to Table 2 as the orders they are displayed in the manuscript.
10) Figure 3 has not sufficient resolution.
11) Figure 6B, how do you define the clinical scores of the infected pigs. Please provide the detailed scoring methods in the material and method section.
12) Figure 7, pleased provide more descriptions on what the lesions are per your arrow pointed sites, especially for the HE straining sections. For the lung sections, I cannot understand why the vacuoles which should be a common micromorphology for lungs were defined as the lesion. In addition, please provide literature references or diagnosis standards for validation of the PRRSV-associated pathological changes.
13) Figure 8B is not informative. What do these colored lines stand for?
14) Figure 8C, how do you measure the food intake? The testing methods should be provided in the material and method section.
15) Figure 8D, please provide scoring standards.
16) For antibody assay between Fig 6G and 8E, different cut lines were used (0 for Fig. 6G and 0.4 for Fig. 8E). Please explain this inconsistency. Additionally, challenged piglets and pregnant sows seem to have differential antibody development trends, please explain.
17) The methods and reagents for assaying antibody titers and viral loads were not included in the manuscript.
Author Response
Dear Reviewer:
Thank you very much for handling our manuscript for review. We have spent a lot of time carefully revising and supplementing your questions. Our point-to-point response to the reviewers is list as following.
Reviewer: 2
This manuscript provides new information concerning the characterization and pathogenicity of 2 PRRSVs recombinant strains. The study seems well-designed and implemented. The authors made sequence analysis and viral challenge experiments to support their view. The study can value the current pig production by deciphering the PRRSV evolution and epidemiology in China.
My major concerns to this work are that 1) the manuscript is poorly prepared. The text lacks clarity in Material and Methods section, Results, Tables and Figure Legends. Moreover, a good English editing is suggested. My view is that the manuscript is not yet ready for publication in its present form. 2) The significance of PRRSV recombination between NADC30 and NADC34 should be analyzed and discussed. Does such recombination has been found elsewhere/ or it is the first report here? How this recombination made the changes to the infectivity or pathogenicity of current prevalent PRRSV strains? How this recombinant strain affects the current pig production and how we should act to the emergence of new recombinant strains?
R: 1) Thank you very much for your careful review of our manuscript and your valuable comments; We have revised and supplemented the manuscript according to your questions, while further editing and revising the English writing.
R: 2) Thank you again for your questions, and we will explain them one by one.
First of all, from the development and epidemic trend of PRRSV in China, the evolution, mutation and recombination of PRRSV strains all seriously endanger the production situation of pig breeding. In addition, from the current treatment and prevention measures of PRRSV, the traditional vaccine has a weak cross-protection power against the virus strains, the early warning and prevention are effective prevention and control measure. Besides, NADC30-like and NADC34-like strains have become the mainstream strains in recent years, so it is very likely that new recombinant strains will occur between the two sublineages, and the virulence of the recombinant strains are very necessary for pig farm prevention.
Second, our study is not the first time to report this kind of recombinant virus. At the beginning of 2022, China Harbin Animal Research Institute indicated that NADC30-like and NADC34-like recombinant strains had occurred in pig farms, but the research on their pathogenicity has not been reported. However, this paper is the first time to study the pathogenicity of such recombinant strains, which provides a more urgent need for production.
Third, the current study shows that NADC30-like strains are less pathogenic to piglets and sows, while NADC34-like strains are less pathogenic to piglets, and the severity of pathogenicity to sows will be related to the strain of recombination. Therefore, whether the recombination of these two strains will enhance their pathogenicity to piglets and sows, it needs to be verified by clinical animal challenge experiment. According to the results of our animal challenge experiments, the recombinant strains were highly pathogenic to sows, and directly leads to the death and abortion of sows, so the epidemic of such recombinant virus may bring serious threat to the pig industry.
Fourth: for the emergence of this recombinant virus, in addition to strengthening biosafety management in production, the development of effective vaccine is also very urgent. Those will also be the focus of our next research, through studies of recombinant viral virulence genes and novel chimeric vaccines by reverse genetics to prevent it.
Furthernore, below the Minor Points that need to be addressed by the Authors.
- Line 88, PrimeScript™ RT-PCR kit does not belong to Promega products.
R: Thank you for pointing out the mistake! We have corrected it in the revised manuscript.
- Lines 99-100, antibody information should be provided.
R: Thank you for the advice! We have added the antibody information in the revised manuscript.
- Line 149, what is “cytoplesions”?
R: Thank you for pointing out the mistake! We have corrected it in the revised manuscript.
- Figure 1A, please describe the detailed macro- or microscopic manifestations of the shown lesion sections, as the specific arrows were used to point to some parts which actually cannot be visually distinguished the adjacent tissues in your images.
R: Thank you for pointing out this! We have corrected it in the revised manuscript. PRRSV infection can cause fever in pigs, leading to obvious congestion and redness in the body epidermis, which is a very obvious onset symptom clinically.
- Figure 1B, what do you mean by using “gel electrophoresis” here? I guess it is a PCR result. I cannot find any information about what gene or sequence you used for amplification and how you determine which samples are positive from this result.
R: Thank you for pointing out the mistake! The PRRSV GP5 gene (603bp) fragment was amplified by PCR to screen for the positive samples. The primer sequences are supplemented in Materials and methods.
6) Figures 1C&D, high resolution images for cell morphology and staining should be provided. The cells and fluorescence shown in fig 1C&D was barely discernible in the reviewed version of manuscript.
R: Thank you for the advice! We re-supplied the high-resolution images in the revised manuscript.
- Figures 1E&F, all data points are not placed at the indicated time points.
R: Thank you for pointing out the mistake! We have corrected it in the revised manuscript.
8) Line 140, please describe what is “diseased material”.
R: Thank you for the suggestion! Actually "diseased material" means tissue samples, we have corrected it in the revised manuscript.
- Table 1 should be placed prior to Table 2 as the orders they are displayed in the manuscript.
R: Thank you for the advice! We have corrected it in the revised manuscript.
- Figure 3 has not sufficient resolution.
R: Thank you for pointing out the mistake! We have corrected it in the revised manuscript.
11) Figure 6B, how do you define the clinical scores of the infected pigs. Please provide the detailed scoring methods in the material and method section.
R: Thank you for pointing out this! We have supplemented the relevant scoring details and criteria in “Materials and methods 2.3”.
- Figure 7, pleased provide more descriptions on what the lesions are per your arrow pointed sites, especially for the HE straining sections. For the lung sections, I cannot understand why the vacuoles which should be a common micromorphology for lungs were defined as the lesion. In addition, please provide literature references or diagnosis standards for validation of the PRRSV-associated pathological changes.
R: Thank you for pointing out the mistake! I am very sorry for my unprofessional statements. PRRSV infection causes lung pathology damage, resulting in inflammatory infiltration, bleeding, consolidation, and interval thickening. As a result, some cell lesions atrophy, or thicken, causing some normal lung vacuoles to become relatively larger, and we have relabeled the pathological lesions.
13) Figure 8B is not informative. What do these colored lines stand for?
R: Thank you for pointing out the mistake! We have corrected it in the revised manuscript.
14) Figure 8C, how do you measure the food intake? The testing methods should be provided in the material and method section.
R: Thank you for the advice! We have corrected it in the revised manuscript. The feed fed to the sows every day is certain. If there is residual feed in the trough , it is cleaned up and weighed to calculate the daily feed intake.
15) Figure 8D, please provide scoring standards.
R: Thank you for pointing out the mistake! We have corrected it in the revised manuscript. The same scoring standards as for piglets.
- For antibody assay between Fig 6G and 8E, different cut lines were used (0 for Fig. 6G and 0.4 for Fig. 8E). Please explain this inconsistency. Additionally, challenged piglets and pregnant sows seem to have differential antibody development trends, please explain.
R: Thank you for pointing out the mistake! We have corrected it in the Fig. 6G.
17) The methods and reagents for assaying antibody titers and viral loads were not included in the manuscript.
R: Thank you for pointing out this! We have corrected it in the revised manuscript.

Round 2
Reviewer 2 Report
Figure 7A – There are 2 types of arrows with different colors, but no explanations for what they stand for. The information of these marks must be clearly annotated in figure legends and the corresponding contents in result section.
For line 288 “in lymphoid histopathological sections shown by the white arrow (Figure 7B)”, the reviewer cannot find the claimed white arrow in Figure 7B.
Line 318- Figure A?
Line 378- Harbin Animal Research?
Lines 400-417, the language use is bad in the labeled paragraph. A serious revision of English is required to improve the language quality, including the whole manuscript.
Author Response
Dear Reviewer:
Thank you very much again for handling our manuscript for review. We have spent a lot of time carefully revising and supplementing your questions. Our point-to-point response the reviewers is list as following.
Reviewer: 2
Figure 7A – There are 2 types of arrows with different colors, but no explanations for what they stand for. The information of these marks must be clearly annotated in figure legends and the corresponding contents in result section.
R: Thank you for pointing out this! We have corrected it in the revised manuscript. In Figure 7A the white arrow indicates the site of tissue bleeding; however, the black arrows in Figure 7B indicates lung lesions in HE straining sections, as well as lymph node bleeding lesions.
For line 288 “in lymphoid histopathological sections shown by the white arrow (Figure 7B)”, the reviewer cannot find the claimed white arrow in Figure 7B.
R: Thank you for pointing out the mistake! We have corrected it in the revised manuscript.
Line 318- Figure A?
R: Thank you for pointing out the mistake! We have corrected it in the revised manuscript. The correct statement is Figure 8A.
Line 378- Harbin Animal Research?
R: Thank you for pointing out the mistake! We have corrected it in the revised manuscript. The correct statement is Harbin veterinary research institue.
Lines 400-417, the language use is bad in the labeled paragraph. A serious revision of English is required to improve the language quality, including the whole manuscript.
R: Thank you for the advice! We have carefully revised this section as well as whole manuscript.